# Low Maternal Immunoglobulin G Avidity and Single Parity as Adverse Implications of Human Cytomegalovirus Vertical Transmission in Pregnant Women with Immunoglobulin M Positivity

**DOI:** 10.3390/v13050866

**Published:** 2021-05-09

**Authors:** Masatoki Kaneko, Junsuke Muraoka, Kazumi Kusumoto, Toshio Minematsu

**Affiliations:** 1Department of Obstetrics and Gynecology, Faculty of Medicine, University of Miyazaki, 5200 Kihara, Kiyotake, Miyazaki 889-1692, Japan; jyunsuke_muraoka@med.miyazaki-u.ac.jp (J.M.); kazumi_kusumoto@med.miyazaki-u.ac.jp (K.K.); 2Graduate School of Nursing Science, Faculty of Medicine, University of Miyazaki, 5200 Kihara, Kiyotake, Miyazaki 889-1692, Japan; 3Research Institute for Disease Control, Aisenkai Nichinan Hospital, 3649-2 Kazeda, Nichinan, Miyazaki 887-0034, Japan; tmine@med.miyazaki-u.ac.jp

**Keywords:** human cytomegalovirus, congenital infection, immunoglobulin g avidity, pregnancy, preventive measure

## Abstract

Human cytomegalovirus (CMV) is the leading cause of neurological sequelae in infants. Understanding the risk factors of primary CMV infection is crucial in establishing preventive strategies. Thus, we conducted a retrospective cohort study to identify risk factors of vertical transmission among pregnant women with immunoglobulin (Ig) M positivity. The study included 456 pregnant women with IgM positivity. Information on age, parity, occupation, clinical signs, IgM levels, and IgG avidity index (AI) was collected. The women were divided into infected and non-infected groups. The two groups showed significant differences in IgM level, IgG AI, number of women with low IgG AI, clinical signs, and number of pregnant women with single parity. In the multiple logistic regression analysis, pregnant women with single parity and low IgG AI were independent predictors. Among 40 women who tested negative for IgG antibody in their previous pregnancy, 20 showed low IgG AI in their current pregnancy. Among the 20 women, 4 had vertical transmission. These results provide better understanding of the risk factors of vertical transmission in pregnant women with IgM positivity.

## 1. Introduction

Human cytomegalovirus (CMV) is the most common virus that causes morbidity and mortality in congenitally infected fetuses and newborns, resulting in a broad range of disabilities, including sensorineural hearing loss, visual impairment, and motor and cognitive deficits [1]. The prevalence rates of congenital CMV infection range from 0.2% to 2.0% in newborns, and 10–15% of newborns with CMV infection are symptomatic [1,2]. In Japan, congenital CMV infection was reported to occur in 0.31% of newborns [3].

No global consensus has been reached regarding maternal serum screening for CMV infection [1,2,4,5]. No screening protocols for detecting congenital infection and no effective measures for intrauterine treatment have been established, and no vaccine has been developed to protect against the infection. Thus, maternal serum screening is considered useful for identifying seronegative women, who are advised on preventive measures such as washing their hands after caring for infants and avoiding contact with children’s body fluids as much as possible [6]. In addition, universal screening would aid in starting a proper ultrasound follow-up of fetuses whose mothers’ results show CMV infection. Knowing of the presence of maternal infection can result in a better interpretation of imaging details that usually are not considered pathological. An example could be abnormality of the fetal heart rate, which can be caused by neurological damage due to CMV infection [7]. 

Congenital CMV infection may result in intrauterine fetal death, neonatal death, intrauterine growth restriction, and preterm birth. Approximately 10% of infected newborns are symptomatic, with findings such as unilateral or bilateral sensorineural hearing loss, vision loss, optic atrophy, strabismus, chorioretinitis, microcephaly, hepatomegaly, splenomegaly, thrombocytopenia, petechiae, jaundice, seizures, and mental disability. In addition, approximately 15% of initially asymptomatic CMV-infected newborns develop long-term neurological sequelae before the age of 5 years [1].

In addition, primary CMV infection was detected in only 25% of CMV IgM-positive pregnant women [8]. Accordingly, if positive results of tests for IgM antibodies are interpreted incorrectly to mean that the infection was recently acquired, this interpretation could influence a pregnant woman’s decision to terminate her pregnancy. A retrospective cohort study conducted in Israel reported that many pregnant women with suspected CMV infection during the first trimester chose to terminate their pregnancy [9]. The authors concluded that lack of information regarding CMV infection during pregnancy may be a factor in the decision to terminate their pregnancy and that correct interpretation and communication of confirmatory test results by expert physicians may significantly reduce the rate of unnecessary termination [9]. In addition, we previously showed that pregnant women with CMV IgM positivity had more severe anxiety levels than those with CMV IgM negativity [10]. Accordingly, maternal mental health care is required when maternal serum screening is introduced.

The IgG avidity assay has some limitations. First, the timing of the assay execution influences the result owing to the progress of IgG maturation over time. A second limitation is that the kinetics of IgG avidity maturation in pregnant women with primary infection differ between vertical and non-vertical transmission cases [11]. However, IgG avidity testing is useful for differentiating between recent primary infection and CMV IgM-positive non-primary infection and false-positive CMV IgM test results, and for identifying pregnant women with a high risk of vertical transmission [1,2,4,5,12,13]. Thus, further examination is required to confirm the usefulness of the IgG avidity assay. 

Recently, more than two-thirds of all congenital CMV infections in the USA were estimated to occur in infants born to women with non-primary maternal CMV infection [14,15]. However, there is no obvious evidence regarding the occurrence rate of congenital CMV infection among seropositive pregnant women in Japan. We showed that only one infected infant was born among 929 seropositive pregnant women in a local area of Japan [16]. The prevalence of congenital CMV infection is influenced by racial, ethnic, and socioeconomic background. 

In addition, clinical symptoms are more likely to be present in women with primary infection than in women with recurrent infections or reactivations [17]. Thus, we believe that pregnant women’s anamnestic data, including known or accidental CMV exposures, and serological test results must be considered in developing preventive strategies against primary CMV infection.

Accordingly, we conducted a retrospective cohort study to identify the clinical risk factors of vertical transmission among pregnant women with CMV IgM positivity.

## 2. Materials and Methods

### 2.1. Patients and Methods

CMV IgM-positive pregnant women who were referred to Miyazaki University Hospital between 2009 and 2019 for clinical consultation were enrolled in this study after obtaining informed consent. Pregnant women with CMV IgM positivity were identified on the basis of maternal serum screening results, measurement of CMV antibodies due to maternal infectious signs, or measurement of CMV antibodies due to abnormal fetal ultrasonography findings.

Maternal serum screening for primary CMV infection was conducted at selected clinics. Blood samples for CMV IgM and IgG analyses were collected at the patients’ expense, simultaneously with routine maternal serum screening at their health checkup. A commercial enzyme immunoassay kit (Denka Seiken, Tokyo, Japan) was used for the measurements. The cut-off CMV IgG and IgM levels were 2.0 and 1.2, respectively. The maternal serum samples were routinely stored in the laboratory center for 1 week in case of clinical requirement.

The same maternal sera used to measure CMV IgG and IgM were used to measure CMV IgG AI at Aisenkai Nichinan Hospital. IgG avidity tests were performed as previously described, with slight modifications. The Enzygnost anti-CMV enzyme-linked immunosorbent assay kit (Siemens Healthcare Diagnostics, Tokyo, Japan) was used for the analysis. Antibody AI (%) was calculated as the mean absorbance at 450 nm (OD450) of urea-washed wells divided by the mean OD450 of the control wells without urea washing. IgG AI values <35%, 30–50%, and ≥50% were defined as low, moderate, and high, respectively, according to previous studies [13,16,18,19].

Maternal information on age, obstetric history, and occupation was obtained from medical charts. The pregnant women participated in an interview about the presence of fever or flu-like symptoms during pregnancy at their first visit to the university hospital.

Amniocentesis at approximately 20 weeks’ gestation was offered to the pregnant women to confirm fetal infection using polymerase chain reaction (PCR). PCR to confirm fetal infection using neonatal urine was performed within 2 weeks for all neonates born to pregnant women who were CMV IgM positive.

The pregnant women were divided into three groups according to IgG AI (low: <35%, moderate: 35–50%, and high: ≥50%) to investigate their characteristics. Nonspecific immunoreaction to CMV IgM was judged when the results were negative for IgG, positive for IgM, and 0 for IgG AI after repeated serological tests at appropriate intervals.

### 2.2. Statistical Analyses

Between-group differences were assessed using the Mann–Whitney *U*-test, χ^2^ analysis, or Fisher exact test. Correlation between the CMV IgG AI and IgM level was assessed using the Spearman rank test. Results with *p* values < 0.05 were considered statistically significant.

A multivariable logistic regression analysis was performed to identify any independent predictive factors of fetal CMV infection. Only predictive variables with a *p* value < 0.1 in the univariate analysis were entered into a logistic regression model. A stepwise forward procedure using the likelihood ratio test was used in the multivariable logistic regression analysis. Variables with *p* values < 0.05 in the final model of the multivariable logistic regression were determined to be independent predictive factors of fetal CMV infection. A statistical analysis was performed using the SPSS version 22 software program for Windows (IBM SPSS Statistics, Tokyo, Japan). Data are presented as mean ± standard deviation.

## 3. Results

Four hundred and fifty-seven pregnant women were referred to the university hospital during the study period. Among them, one was excluded because of cessation of pregnancy after judgment of CMV IgM positivity. As a result, 456 pregnant women were enrolled in the study. The numbers of pregnant women with low, moderate, and high IgG AI were 83, 89, and 284, respectively. Their characteristics are shown in Table 1. The proportion of primiparity among the pregnant women with low IgG AI was significantly lower than that among the pregnant women with high IgG AI (*p* < 0.01). The proportion of single parity among the pregnant women with low IgG AI was significantly higher than that among the pregnant women with high IgG AI. Congenital CMV infection was confirmed in 19 cases, including 13 (15.7%) with low IgG AI, 0 with moderate IgG AI, and 6 (2.1%) with high IgG AI. A statistically significant difference in the number of infected cases was found between the low and high IgG AI groups (*p* < 0.01). We found no significant differences in the timings of the measurements of IgM level and IgG AI between the three IgG AI groups. The CMV IgM level in the low IgG AI group was higher than that in the other two groups (*p* < 0.01).

Among the pregnant women with low IgG AI, four (4.8%) had an IgG AI of 0 throughout their pregnancy in spite of CMV IgM positivity and confirmed nonspecific immunoreaction to CMV IgM. Four women showed seroconversion during pregnancy, including three in the low IgG AI group and one in the high IgG AI group. In the three pregnant women with low IgG AI, the IgG AI values (gestational weeks of testing) were 3.4 (29), 7.1 (28), and 27.6 (38), respectively. Among these women, those with an IgG AI of 27.6 had vertical transmission. On the other hand, one pregnant woman with a high IgG AI had an IgG AI of 73.2 at 36 weeks of gestation and vertical transmission (Table 1). There were no pregnant women with risk factors for pre-conception immune suppression.

Table 2 shows the characteristics of infected cases. Thirteen pregnant women had low IgG AI. Among them, one chose to terminate her pregnancy, and four had infants with neurological sequelae. Six pregnant women had high IgG AI, and two showed high IgG AI during the first trimester.

Among this study population, 77 pregnant women were tested for both IgG and IgM antibodies in their previous pregnancy, and 40 tested negative for these antibodies at that time (Table 3). Among the 40 women, 20 (50%) showed low IgG AI in their current pregnancy, of whom 4 had vertical transmission (Table 3). The frequency of para 1, para 2, and para 3 pregnant women among the 40 pregnant women with antibody negativity in their previous pregnancy were 23 (57.5%), 14 (35%), and 3 (7.5%), respectively. Thirty-seven pregnant women had both IgG and IgM positivity in their previous pregnancy and remained positive for IgG and IgM in their current pregnancy (Table 3). Among them, one pregnant woman showed low IgG AI (19.5%) in her previous pregnancy, which remained low (22.6%) in her current pregnancy (IgG: 4.5, IgM: 3.89).

The study subjects were divided into the infected and non-infected groups (Table 4). We found no significant difference in maternal age between the two groups. The timing of the serological tests in the infection group was later than that in the non-infection group (*p* < 0.05). The CMV IgM value in the infected group was significantly higher than that in the non-infected group (*p* < 0.01). The CMV IgG AI in the infected group was lower than that in the non-infected group (*p* < 0.01). A negative correlation was observed between the CMV IgM values and the IgG AI (*r* = −0.363, *p* < 0.01). Significant differences were observed in the proportions of pregnant women with low IgG AI, para 1, and pregnant women with maternal fever or flu-like symptoms between the two groups (*p* < 0.05).

The findings from the multiple logistic regression analysis are shown in Table 5. Four predictive factors, namely maternal age, single parity, maternal fever or flu-like symptoms during pregnancy, and low IgG AI during pregnancy, were entered into the multivariate model. As a result, two predictive factors, namely single parity and low IgG AI during pregnancy, were found to be independent predictive factors of congenital infection.

## 4. Discussion

Our results indicate that low IgG AI during pregnancy was the most profound risk factor of congenital CMV infection in pregnant women with CMV IgM positivity. This result corresponds with those in the previous reports [1,2,4,5,12,13]. However, we indicated some limitations of IgG AI as a predictive factor in this study. First, the timing of the assay execution influences the IgG AI. In our study, when we compared the timing of assay execution between the pregnant women with high and those with low IgG AI among the transmission cases, the assay timing tended to be later in the pregnant women with high IgG AI, though without a statistically significant difference. Moreover, one pregnant woman with vertical transmission who seroconverted showed a high IgG AI at 36 weeks of gestation. IgG avidity maturation progresses over time. The kinetics of IgG avidity maturation in primary infected pregnant women was also reported to show different patterns according to duration and intensity [11]. Therefore, the timing of the IgG avidity assay must be considered when using its value in the judgment of primary maternal infection. We previously showed that a low IgG AI with IgM positivity at ≤14 weeks of gestation was a good indicator of congenital infection as compared with that at ≥15 weeks of gestation [13]. Revello et al. showed that a high IgG AI detected in the first trimester has a negative predictive value of 100% for determining the risk of vertical transmission, whereas intermediate-to-high values obtained after 21 weeks of gestation cannot rule out a primary infection (with the negative predictive value decreasing to 91%) [17]. Second, one pregnant woman with IgM positivity showed a long persistence (>2 years) of low IgG AI. This may potentially result in the misdiagnosis of primary CMV infection, particularly when CMV IgM antibodies are detected. Although the reason for this phenomenon is unclear, Lumley et al. also observed long persistence of >18 weeks of low IgG AI by using an Abbott Architect assay [20]. Finally, we observed an IgG AI of 0 in the pregnant women with IgM positivity and judged a nonspecific immunoreaction to CMV IgM after repeated measurements of IgG level, IgG AI, and IgM level. This may also potentially result in the misdiagnosis of primary CMV infection.

Single parity was also a risk factor of congenital CMV infection in pregnant women with CMV IgM positivity. We believe that there are some clinical conditions in pregnant women related with vertical transmission; maternal seronegativity in the periconceptional period or during the pregnancy, presence of infection origin, exposure period with infection origin. It appears that the infected toddlers are more likely the source of infection. This is because although most infected toddlers were asymptomatic, they tended to shed large amounts of virus in their saliva or urine after their first infection [1,21]. This is the reason why exposure to infected toddlers is the most common means of contracting CMV infection among pregnant women. Exposure to CMV among women likely increases with an increase in the number of their children. Accordingly, a strong supposition is that pregnant women with ≥2 children have many opportunities for exposure to CMV before conception and thus become seropositive in their current pregnancies. On the other hand, primiparous women likely have few opportunities for exposure to the virus during pregnancy or before conception. Thus, we believe that women pregnant with their first child have a higher risk of primary infection during pregnancy. However, our second analysis may reveal another reason. We also examined 40 pregnant women who were seronegative in their previous pregnancy and found that approximately 58% of the women were para 1 and 35% were para 2. This result might show the importance of intervals between consecutive births or maternal immunity to CMV infection, rather than the number of parities, in the risk of transmission in utero. Regarding intervals between consecutive births, Fowler et al. reported that the risk of transmission in utero was highest among mothers who delivered ≤24 months after their previous delivery and mothers who seroconverted between deliveries [22]. We were unable to evaluate the intervals between consecutive births as a risk factor in the present study because the exact interval was unclear. This is a limitation of our study. The parity factor is not useful for maternal screening for congenital CMV infection. However, we believe that the parity factor is meaningful when considered as a preventive measure for vertical transmission, i.e., pregnant women who are seronegative for CMV, with short intervals between consecutive births, or with single parity pay more attention to hygiene measures. The development of effective vaccination or other effective treatment to prevent vertical transmission is expected. There are several studies with respect to the effectiveness of hyperimmune globulin (HIG) to prevent vertical transmission. Revello et al. conducted a randomized trial and found that this treatment did not significantly modify the course of primary CMV infection during pregnancy [23]. However, Kagan et al. reported that HIG was effective for women with a recent primary infection in the first trimester or during the periconceptional period and when HIG was administered at a biweekly dose of 200 IU/kg [24].

Childcare workers are potentially at risk of occupational exposure to the CMV, the leading cause of congenital infection [25]. However, no statistically significant difference in the number of childcare staff was found between the infected and non-infected groups in this study. Accordingly, this variable was excluded from the multiple regression analysis. However, it is rash to conclude only from our result that childcare staff is not at risk of primary infection. Several factors, including shedding levels in day care centers, capacity of day care centers, and contact period with children aged <3 years, have been reported to influence the occurrence of primary infection [25,26,27].

Maternal fever or flu-like symptoms were not an independent risk factor in the multiple regression analysis, although a significant difference in this factor was found between the infected and non-infected groups. Maternal primary infection is difficult to diagnosis on the basis of clinical symptoms alone, as these are nonspecific, and 25–50% of mothers are asymptomatic [14,28]. In addition, maternal symptoms might include those induced by another specific infection due to the fact that the information was collected retrospectively.

There were no pregnant women with risk factors for pre-conception immune suppression in this study. However, this factor is important when considering vertical transmission. Several studies recently assessed the role of CMV cell-mediated immunities in pregnant women. The delayed development of the memory CD4^+^ T-cell response was associated with a higher risk for vertical transmission in pregnant women after primary CMV infection [29]. Conversely, Saldan et al. suggested that high cell-mediated immune responses promote CMV vertical transmission in primary infection, whereas preexisting cell-mediated immunity in non-primary infection exerts protective effects against fetal infection [30]. In addition, Cavoretto et al. reported that CMV-seropositive pregnant women receiving immunosuppressive agent showed a severe reduction in both total and HCMV-specific CD4^+^ T cells and delivered severely symptomatic newborn [31]. Thus, although several studies have suggested that HCMV-specific, T-cell-mediated immune responses play a crucial role in controlling vertical transmission, further study is required regarding the specific features of responses that contribute to protection against vertical transmission.

The rate of vertical transmission among pregnant women with IgM positivity was 0.04% in this study population. Among infected cases, two pregnant women may have been suspected non-primary infection, regardless of maternal IgM positivity, because they had high IgG AI at the beginning of the first trimester. The incidence of pregnant women with low IgG AI among those with IgM positivity was 18.2%. This does not indicate the incidence of primary infection because not all pregnant women had IgG AI measured during the first trimester. We conducted a cohort study of 1163 pregnant women in 2008, measuring IgG and IgM antibodies at 10.8 ± 2.2 weeks of gestation [16]. Forty pregnant women (3.4%) had IgM positivity and nine among them had low IgG AI. The incidence of primary infection during the first trimester was 0.77% [16]. 

## 5. Conclusions

The rate of vertical transmission among pregnant women with IgM positivity was 0.04% in this study population. The most profound predictive factor for CMV vertical transmission was low IgG AI. Among pregnant women with negative CMV IgG antibody in the previous pregnancy, 50% showed low IgG AI and 20% among them had CMV vertical transmission in the present pregnancy.

## Figures and Tables

**Table 1 viruses-13-00866-t001:** Characteristics of the pregnant women according to IgG avidity index.

		IgG Avidity Index		
	Total (n = 456)	Low (n = 83)	Moderate (n = 89)	High (n = 284)
Age (years)	29.7 ± 4.7	28.8 ± 4.7	29.8 ± 4.3	29.9 ± 4.8
Parity				
0	155 (34.0%)	19 (22.9%) *	27 (30.3%)	109 (38.4%)
1	169 (37.1%)	41 (49.4%) *	31 (34.8%)	97 (34.1%)
≥2	132 (28.9%)	23 (27.7%)	31 (34.8%)	78 (27.5%)
Infection	19 (4.2%)	13 (15.7%) *	0	6 (2.1%)
Seroconversion during preg.	4 (0.9%), (2)	3 (3.6%), (1)	0	1 (0.4%), (1)
amniocentesis	24 (5.3%), (6)	11 (13.3%), (4)	2 (2.2%), (0)	11 (3.9%), (2)
G.W. at measurement	12.5 ± 4.1	13.3 ± 5.6	12.1 ± 3.1	12.4 ± 3.7
CMV IgM	2.85 ± 1.79	4.34 ± 2.51 *	2.66 ± 1.53 **	2.47 ± 1.35
IgG avidity index	53.9 ± 20.6	20.0 ± 10.5 *	43.8 ± 4.5 **	66.9 ± 10.3 ***

(); the number of infections, Abbreviations: pre, previous; preg, pregnancy; G.W., gestational weeks; * *p* < 0.05, low versus high IgG AI; ** *p* < 0.05, low versus moderate IgG AI; *** *p* < 0.05, moderate versus high IgG AI.

**Table 2 viruses-13-00866-t002:** Characteristics of CMV-infected cases.

				IgG Avidity Assay		Neonate		
No.	Parity	Maternal Sign	IgG Antibody in Previous Preg.	Timing (weeks)	IgG AI	G.W. at Birth	B.W. (g)	Prognosis
1	3	No	Negative	12	1	38	2866	Normal
2	1	No	Unknown	17	3	40	3296	Normal
3	1	Yes	Unknown	14	3.1	39	3048	Normal
4	1	No	Unknown	12	6.6	40	2796	Normal
5	2	No	Negative	10	8.5	39	2756	Normal
6	1	No	Unknown	12	9.7	37	2604	bil. SNHL, MR
7	1	No	Unknown	11	15.9	Termination	Termination	Hydrops
8	1	No	Unknown	13	19.8	38	3194	Normal
9	1	No	Negative	10	21.9	39	2685	Normal
10	1	Yes	Negative	38	27.6	39	3042	Normal
11	0	No	Unknown	21	29.3	37	2176	bil. SNHL, MR
12	0	No	Unknown	26	30	29	870	bil. SNHL, MR
13	1	Yes	Unknown	11	31.7	38	2830	bil. SNHL, MR
14	0	Yes	Unknown	23	69.5	38	2296	bil. SNHL, MR
15	1	No	Unknown	8	69.9	40	3100	Normal
16	1	No	Negative	36	73.2	36	1683	Normal
17	1	No	Unknown	37	79.9	37	2510	Normal
18	1	Yes	Unknown	7	80.7	38	2890	Normal
19	0	No	Unknown	24	89.5	28	754	bil. SNHL, MR

AI, avidity index; G.W., gestational week; B.W., birth weight; bil, bilateral; SNHL, sensory nerve hearing loss; MR, mental retardation; preg, pregnancy.

**Table 3 viruses-13-00866-t003:** IgG antibody positivity in the previous pregnancy according to IgG AI.

IgG avidity in the Present Pregnancy		IgG in the Previous Pregnancy	
	*n*	Negative	Positive *
Low	21	20 (4)	1
Moderate	18	11 (0)	7
High	38	9 (1)	29
Total	77	40 (5)	37

* These women also showed IgM positivity in their previous pregnancy. Abbreviations: AI, avidity index; Ig, immunoglobulin (): the number of infections.

**Table 4 viruses-13-00866-t004:** Comparison between the infection and non-infection groups.

	Infection	Non-Infection	Statistics
	*n* = 19	*n* = 437	
Age (years)	28.8 ± 4.8	29.7 ± 4.7	0.067
Timing of serological tests (weeks)	18 ± 10	12.3 ± 3.4	<0.05
CMV IgM value	4.6 ± 3.0	2.8 ± 1.7	0.003
IgG avidity index	35.3 ± 30.9	54.7 ± 19.7	0.006
Pregnant women with low AI	13 (68.4%)	70 (16.0%)	<0.01
Pregnant women with single parity	13 (68.4%)	156 (35.7%)	0.006
Childcare staff	1 (5.6%)	25 (5.7%)	1.0
Maternal fever or flu-like symptoms	5 (27.8%)	33 (7.5%)	0.015

Data are shown as mean ± standard deviation or *n* (%). Abbreviations: CMV, cytomegalovirus; Ig, immunoglobulin; AI, avidity index.

**Table 5 viruses-13-00866-t005:** Results of the multivariable logistic regression analysis of the predictive factors.

Factors of Vertical Transmission					
				95% CI	
Variable	B	*p* value	OR	Lower	Upper
Low AI	2.3	<0.01	9.98	3.63	27.45
Pregnant women with single parity	1.12	0.031	3.08	1.11	8.54
Constant	1.22	<0.01			

Abbreviation; AI, avidity index; OR, odds ratio; CI, confidence interval.

## Data Availability

The data presented in this study are available on request from the corresponding author. The data are not publicly available due to restrictions for personal information protection.

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
