# Peer review of "Low Maternal Immunoglobulin G Avidity and Single Parity as Adverse Implications of Human Cytomegalovirus Vertical Transmission in Pregnant Women with Immunoglobulin M Positivity"

_viruses, 2021, doi:10.3390/v13050866_

Round 1

Reviewer 1 Report

This article by Kaneko and colleagues described a cohort of 457 pregnant women with IgM positivity and different IgG AI for their risk of passing virus to their fetuses. The study is a retrospective study and constrained to a local hospital based on referrals. Nonetheless, the study is interesting in revealing the importance of congenital CMV in Japan. The finding of low IgG AI is correlated with congenital transmission risk is not entirely novel, although important to demonstrate in Japanese community for future awareness. The real world data on Japanese population would be a powerful way to advocate the urgency of this medical problem to government and public health officials. This reviewer (US based) was a little surprised to the fact as mentioned in the manuscript that women have to pay out of their pocket for CMV serology screening during pregnancy. The manuscript could be improved with these considerations 1. It is not sure why authors chose to have three AI groups, esp. the middle one is so small. It may be easier to calculate and analyze two groups of low (below 50%). vs. high (greater or equal 50%) 2. It would a lot more meaningful to get some population statistics from 2009-2019 for the regions from which these women were referred. The readers would appreciate the information of general incidence rate of primary infection in women during pregnancy in this region (hopefully reflect the general population) 3. Although multivariant model is convenient, it is not entirely helpful for forming future screening strategy. based on data, it seems that IgG AI is a very clear determinant for the risk; adding parity factor seemed confusing. It may be helpful in discussion but not that much in conclusion.

Reviewer 2 Report

This is a scientifically sound study assessing the risk of fetal infection in a cohort of 456 patients with CMV IgM positivity. The results are somewhat as expected, neither surprising nor novel. However, they may be useful to the scientific community after correcting some weaknesses. These are my construictive criticism for the authors:

1. Did the authors consider immunosuppression and immune factors among covariates of their model? This may be relevant for the scopes fo their study (ef use of corticosteroids, immunosuppression etc). Previous data showed that pregnant women receiving immunosuppressive treatment may have an increased risk of CMV primary or secondary infection and transmission to the fetus. A recent report showed that a severe reduction in both total and HCMV-specific CD4+ T cells may have the potential to promote maternal reactivation/reinfection with CMV secondary infection, vertical transmission to the fetus and  sequelae. Consider citing this reference and add a brief discussion of this issue. (1)

2. The concept of immune treatment with immunoglobulin is emerging. Despite a recent RCT failed to show utility o f specific hyperimmune Ig treatment (2) a recent observational study showed robust evidence of efficacy to prevent maternal–fetal transmission in pregnancy with a recent primary CMV infection particularly in the first trimester or during the periconceptional period at a biweekly dose of 200 IU/kg (3). Please discuss briefly this issue.

3. The conclusions are not aligned with the results. Please stick to the actual results of the study and do not provide suggestion which are not bases upon the study results. In the conclusions I see a sentence like this: "rate of vertical transmission of mother with IgM positive is about 14/456 (0.031). Significant predictors are primiparity and low IgG avidity."

4. Was there any suspected non primary infection? I guess there were not. Add this information, please. 

(1) Cavoretto, P.I.; et al . Prenatal Management of Congenital Human Cytomegalovirus Infection in Seropositive Pregnant Patients Treated with Azathioprine. Diagnostics 202010, 542. https://doi.org/10.3390/diagnostics10080542

(2) Revello MGLazzarotto TGuerra BSpinillo AFerrazzi EKustermann AGuaschino SVergani PTodros TFrusca TArossa AFurione MRognoni VRizzo NGabrielli LKlersy CGerna GCHIP Study GroupA randomized trial of hyperimmune globulin to prevent congenital cytomegalovirusN Engl J Med 20143701316– 1326.

(3) Kagan, K.O., Enders, M., Hoopmann, M., Geipel, A., Simonini, C., Berg, C., Gottschalk, I., Faschingbauer, F., Schneider, M.O., Ganzenmueller, T. and Hamprecht, K. (2021), Outcome of pregnancies with recent primary cytomegalovirus infection in first trimester treated with hyperimmunoglobulin: observational study. Ultrasound Obstet Gynecol, 57: 560-567. https://doi.org/10.1002/uog.23596

Reviewer 3 Report

In this paper, the authors have retrospectively examined a cohort of 456 pregnant women with IgM positivity for human cytomegalovirus (HCMV), to identify risk factors of vertical transmission. In particular, they performed statistical analyses using data about age, parity, occupation, clinical signs, IgM levels, and IgG avidity index.

Overall, the experimental approach seems correct, and the analyses look right. The topic is relevant and a deep investigation in this field is absolutely deserved. The manuscript is well-written and organized. The figures are adequate.

An important issue that would significantly improve the research is to include information about the newborns; if possible, a paragraph and a table dedicated to the features of the newborns, at least indicating if they are symptomatic or asymptomatic, could be included.

Below some minor improvements to allow the publication of the work:

  • Title: specify “human” cytomegalovirus
  • Lines 21-22: the statistical methods employed in the study should be removed from the abstract and described in the Methods section.
  • Lines 45-47: dietary practices are not specifically indicated as control measures for seronegative women. The only rule regarding food is not to share dishes with small children at home. The sentence can be modified as follow: "avoid as much as possible the contact with children's body fluids".
  • Lines 47-50: the abnormal heart rate pattern is not one of the main elements that are used for the in-utero diagnosis of fetal HCMV infection, even though it is useful. In my opinion, the sentence lacks a link to what was previously said about screening; you can change the paragraph in this way: "Moreover, universal screening would help to start a proper ultrasound follow up of the fetuses whose mother result infected by HCMV. Knowing the presence of the maternal infection can lead to a better interpretation of imaging details that usually are not considered pathological. An example could be the abnormality of heart rate, which can be caused by neurological damage due to HCMV infection"
  • Lines 47-49: the description of clinical settings of congenitally infected fetuses should be expanded
  • Lines 50-54: the detail of maternal anxiety, like that of abnormal heart rate, is not a relevant element in the decision to implement screening for congenital infection. The mention that the presence of positive serology could lead to early termination of pregnancies may be an interesting topic, but it is not properly introduced. I would suggest more emphasis on counseling services to mothers who undertake serology, both in terms of mental health and in terms of the choice to continue or not the pregnancy. Maybe it would be better to highlight this point at the end of the paragraph, the explanation of the second part of serology screening.
  • Lines 59-68: this paragraph is a bit confusing. In my opinion, the importance of IgG avidity has not been fully expressed. In addition, it is not emphasized the importance of secondary infections as a source of serious morbidity for the fetus and how they can generate, albeit less frequently, a symptomatic picture similar to that of primary infection. In addition, I would have moved what was said from line 61 to line 68 to the beginning of the paragraph.
  • Line 74: specify the name of the University Hospital
  • Line 224 asymptomatic instead of asymptotic
  • Lines 188-200: the correlation between single parity and an increased risk of infection is a quite surprising result, since most of the works in the literature indicate as at greater risk seronegative women in their second pregnancy, with a small child at home. I think that a deep discussion of this point is required.
  • Line 254: conclusions should be more comprehensive and detailed.

Round 2

Reviewer 2 Report

The authors appropriately addressed all criticisms raised in the first round of rewiew and the paper is now suitable for publication.